# FREQUENCY-DECOUPLED CROSS-MODAL KNOWLEDGE DISTILLATION

## ABSTRACT

Knowledge distillation (KD) has proven highly effective for compressing large models and enhancing the performance of smaller ones. However, its effectiveness diminishes in cross-modal scenarios, such as vision-to-language distillation, where inconsistencies in representation across modalities lead to difficult knowledge transfer. To address this challenge, we propose frequency-decoupled cross-modal knowledge distillation, a method designed to decouple and balance knowledge transfer across modalities by leveraging frequency-domain features. We observe that low-frequency features tend to capture modality-agnostic, generalizable information, while high-frequency features are more modality-specific. Accordingly, we apply distinct losses to these features: enforcing strong alignment in the low-frequency domain and introducing relaxed alignment for high-frequency features. Additionally, we propose a scale consistency loss to address distributional shifts between modalities, and employ a shared classifier to unify feature spaces. Extensive experiments across multiple benchmark datasets show that our method substantially outperforms traditional KD and state-of-the-art cross-modal KD approaches.

## 1 INTRODUCTION

Knowledge Distillation (KD) has emerged as a fundamental technique for model compression and performance improvement. The core concept of KD involves utilizing a large and high-capacity teacher model to mentor a smaller yet more efficient student model. Through this process, the student model learns to approximate the behavior of the teacher model, often achieving comparable or even superior performance despite its reduced complexity.

Despite the substantial success of traditional distillation methods in unimodal settings, such as image or text tasks, many real-world applications inherently involve multimodal data, including vision, language, and audio. In these cross-modal scenarios, effectively transferring knowledge among modalities presents unique challenges. As a result, researchers have increasingly turned their attention to devise cross-modal knowledge distillation (CMKD) framework to enhance the performance of a student model in one modality by leveraging the knowledge of a teacher model in a different modality.

While preliminary progress has been made in cross-modal distillation, existing methods (Gupta et al., 2016; Thoker & Gall, 2019; Afouras et al., 2020; Liu et al., 2023; Jin et al., 2023) often suffer from several limitations: they are typically restricted to specific scenarios or primarily focus on distillation from a stronger modality to a weaker one. Recently, C2KD (Huo et al., 2024) introduced a cross-modal distillation technique based on logits to reduce the gap between modalities. However, this method overlooks the distillation of challenging samples with misaligned soft labels, which are crucial for effective cross-modal knowledge transfer. Furthermore, C2KD exclusively emphasizes logit-level distillation, while intermediate features, which encapsulate richer modal details and semantic information in cross-modal settings, play a more pivotal role in facilitating complementary and effective knowledge transfer. This motivates us to explore cross-modal feature distillation, addressing the challenges posed by discrepant feature representations and misaligned feature spaces.

In this paper, we thoroughly analyze the feature representations across different modalities and reveal that these features contain both modality-specific and modality-generic information. Inspired

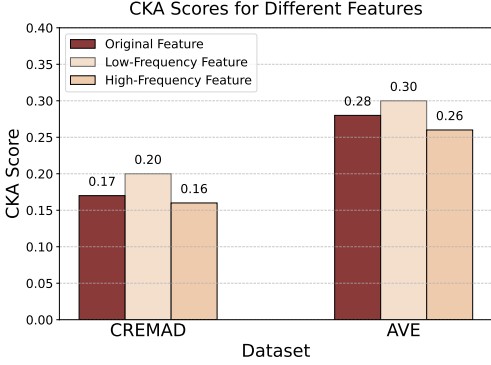 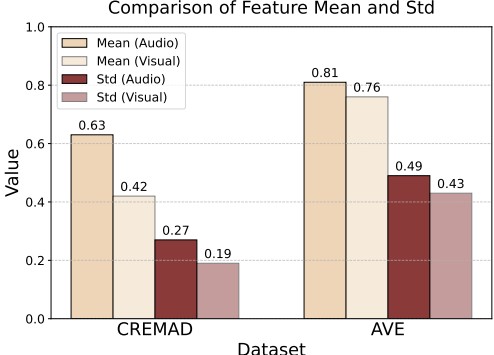

Figure 1: Comparison of CKA scores for different types of features.

Figure 2: Comparison of feature value scale differences across modalities.

by (Williams & Li, 2018; Xu et al., 2020; Pham et al., 2024; Zhang et al., 2024), we investigate the frequency representations of multimodal features, and show that features decomposed into different frequency bands exhibit varying levels of effectiveness for representation. Specifically, *low-frequency features tend to carry more modality-generic information, whereas high-frequency features are more modality-specific.* As illustrated in Figure 1, low-frequency features obtain the highest CKA scores, indicating that they contain more modality-generic information. In contrast, high-frequency features show lower CKA scores, suggesting they are more modality-specific. This sheds light on decoupling and handle these two types of information independently for an effective distillation strategy.

Based on these insights, we propose a new approach of decomposing features into low-frequency and high-frequency components for distillation, applying distinct loss functions to each of them, respectively. For low-frequency features, we employ the traditional mean squared error (MSE) loss to ensure "strong consistency" such that modality-generic information from the teacher model can be better captured. Meanwhile, since high-frequency features contain modality-specific knowledge and exhibit greater variation, making the full alignment less suited, we introduce the logarithmic mean squared error (logMSE) loss to maintain "weak consistency". Furthermore, given that distribution differences are critical for effective knowledge transfer (Pan & Yang, 2009; Li et al., 2019; Sun & Saenko, 2016), we propose a scale consistency loss by the alignment of different modalities through feature standardization, to address the significant discrepancies between modalities. This allows the model to focus on intrinsic discriminative features and reduces the impact of scale variations. We also introduce a shared classifier to align feature spaces further to ensure consistent decision boundaries across modalities, thereby enhancing the effectiveness of cross-modal distillation.

To summarize, our contributions are three-fold:

- By analyzing features across different modalities, we found that low-frequency features exhibit stronger correlations than high-frequency features. Based on this discovery, we introduce a novel cross-modal feature distillation framework. Our method decouples modality-specific and modality-generic information from a frequency perspective, allowing for more effective knowledge transfer by processing each component independently.

- We propose various modality-specific and modality-generic improvements for feature distillation, which enhance focus on valuable discriminative features and improve the alignment of feature representations across different modalities.

- We perform extensive experiments on diverse datasets, covering various modalities and tasks and employing different network architectures to demonstrate the effectiveness of our approach.

## 2 RELATED WORK

### 2.1 GENERIC KNOWLEDGE DISTILLATION

Traditional knowledge distillation methods fall into two main categories: **logit-based distillation** and **feature-based distillation**. Logit-based distillation, first introduced by (Hinton et al., 2015),

transfers knowledge by minimizing the Kullback-Leibler (KL) divergence between the teacher and student model outputs, helping the student learn inter-class relationships. DKD (Zhao et al., 2022) refines this process by separating target and non-target class distillation, allowing better learning of category-specific information. DML (Zhang et al., 2018) enhances transfer by having two models train each other as teachers, while DIST (Huang et al., 2022) uses correlation loss to improve logit distillation by capturing inter-class and intra-class relationships. Feature-based distillation uses intermediate features from the teacher model to supervise the student, aiding in better data representation. FitNet (Romero et al., 2014) was the first to have the student mimic the teacher's intermediate features. Review (Chen et al., 2021) introduced a mechanism that allows the student to learn teacher features layer by layer. Relational Knowledge Distillation (RKD) (Park et al., 2019) focuses on transferring relationships between samples, while PKD (Cao et al., 2022) preserves relational information using Pearson correlation. Attention-based Feature Distillation (AFD) (Ji et al., 2021) utilizes attention mechanisms to dynamically match features, selectively transferring important information. OFD (Heo et al., 2019) employs partial L2 loss, ignoring unhelpful features and focusing on beneficial ones. Generic Knowledge Distillation performs well in single-modal tasks, but due to differences in modality representation, it underperforms in cross-modal scenarios.

## 2.2 Cross-modal Knowledge Distillation

In cross-modal knowledge distillation (CMKD) research, various studies have explored effective methods for transferring knowledge across modalities. (Gupta et al., 2016) proposed an early CMKD framework that transferred labeled supervision from RGB images to depth and optical flow, enhancing the performance of these unlabeled modalities. For action recognition, studies such as (Thoker & Gall, 2019; Dai et al., 2021; Lee et al., 2023) leveraged RGB or optical flow to design CMKD frameworks that improved action detection accuracy. In medical image segmentation, (Wang et al., 2023) addressed missing modalities by selecting the most contributive one for cross-modal distillation in multi-modal learning. CMKD has also been applied to tasks like camera-radar object detection and visual place recognition, as seen in works like (Zhao et al., 2024; Wang et al., 2024). These works are limited to specific scenarios or focus on distillation for individual modalities. (Xue et al., 2022) introduced the Modality Focusing Hypothesis (MFH), offering the first theoretical analysis of CMKD's effectiveness, highlighting modality-generic decisive features as crucial for knowledge transfer. More recently, (Huo et al., 2024) identified modality imbalance and soft label misalignment as major challenges for CMKD, and introduced the C2KD framework, which significantly improved performance through bidirectional distillation and dynamic selection. These works fall short in addressing the inconsistencies in specific modality information and fail to fully leverage modality-generic features for effective cross-modal transfer. Our work overcomes these limitations by introducing a frequency-domain feature decoupling approach.

## 3 How Cross-Modal Features Differ?

The major difference between conventional KD in single modality and cross-modal knowledge distillation (CMKD) is that, CMKD is designed to distill the knowledge from another different modality. This difference poses a significant challenge since the teacher and student are trained with data in different modalities, and therefore have more distinct feature representations. Therefore, to design our cross-modal feature distillation method, it is necessary to first analyze the difference of cross-modal features. In this section, we present two of our major findings: (1) modality-generic and modality-specific features act differently in frequency domain; (2) The features of different modalities exhibit significant differences in scale.

### 3.1 Decoupling Modality-specific and Modality-generic Features

Features from a modality can generally be divided into modality-specific and modality-generic components (Ngiam et al., 2011). The modality-generic component shares the same semantic efficacy across different modalities. While the modality-specific component contains unique and intrinsic information in a certain modality that can hardly be transferred to other modalities. Therefore, to improve knowledge transfer in CMKD, separating these two types of features is crucial. One effective approach to achieving this separation is by analyzing features in the frequency domain (Bruna & Mallat, 2013). By decomposing features into high-frequency and low-frequency components,

we can further decouple the modality-specific and modality-generic aspects of the features. Low-frequency components typically capture the broader, structural patterns that are more likely to be modality-generic and transferable across domains. In contrast, high-frequency components tend to represent finer, modality-specific details that may not generalize as well across modalities.

To demonstrate the above hypotheses, we leverage Centered Kernel Alignment (CKA) (Kornblith et al., 2019) to measure the similarities between different modalities in the decoupled low-frequency and high-frequency features. The calculation method for the CKA Score is in Appendix B. We decoupled the original features of audio and vision models trained on CREMA-D (Cao et al., 2014) and AVE (Tian et al., 2018) datasets, and calculated their CKA scores between audio and vision modalities. As shown in Figure 1, despite all scores are small due to modality and model differences, the low-frequency obtains the highest scores, while the high-frequency are more dissimilar than the original features. This indicates that, more modality-generic information is represented in low frequency, and modality-specific information is more likely to be in high frequency.

As a result, for a better cross-modal distillation, it is beneficial to decouple the features into low-frequency and high-frequency features, and treat them respectively based on their unique characteristics. We will formally introduce our strategy in Section 4.

### 3.2 Scale Differences Across Modalities

In transfer learning, distribution differences play a crucial role in determining whether knowledge can be effectively transferred. If there are significant differences between the distributions of the source and target domains, the model may fail to capture useful information during the transfer process, leading to a substantial decrease in the effectiveness of the transfer.

In cross-modal knowledge distillation, we found that directly using MSE loss based on the raw features resulted in a significant performance drop in the distillation model compared to the single-modal model, as shown in Table 1. For example, the accuracy of the model distilled using raw features on the CREMA-D dataset decreased by 1.5% and 2.5% on the audio and visual modalities, respectively, compared to the uni-modal model. We hypothesize that this sharp decline in performance is likely due to the distribution differences between features from different modalities. We calculated the mean and standard deviation of audio and visual features in the CREMA-D and AVE datasets, as shown in Figure 2. The results show that there are significant differences in the mean and standard deviation between different modalities. For example, on the CREMA-D dataset, the mean and standard deviation of the audio modality features are higher than those of the visual modality by 0.21 and 0.08, respectively.

When MSE is used to force the alignment of the student model's features with those of the teacher model, the student's features may shift towards the mean of the teacher model's features. However, this may conflict with the optimal mean expected in the student's modality, leading to suboptimal performance in the student model.

Therefore, we should not directly use MSE loss to learn features from different modalities. Instead, we should design a loss function that respects the inherent scale differences between modalities to achieve more effective knowledge transfer.

### 4 Our Approach

As previously discussed, we found that modality-specific and modality-generic information can be effectively decoupled through frequency domain analysis, and there are significant differences in the feature distributions across different modalities. In this section, we will formally introduce our method to improve CMKD: (1) We decouple the features into low-frequency and high-frequency components and apply different loss functions for distillation accordingly. (2) We ensure the features from different modalities are consistent in scale and feature space.

### 4.1 Frequency-Decoupled Distillation

We identified frequency decoupling of features as a effective way to disentangle the modality-generic and modality-specific information in the features. Formally, given the original feature $\mathbf{X}^m \in \mathbb{R}^D$

Figure 3: **Framework of our method.** We decouple the features of different modalities in the frequency domain into high-frequency and low-frequency components. For low-frequency features, MSE loss is applied, while logMSE loss is used for high-frequency features. Additionally, we ensure consistency in feature scale and feature space across modalities through feature normalization and alignment modules.

for a certain modality $m$, we compose the following three computation steps to decouple it into two features, namely low-frequency feature $\mathbf{X}_{\text{low}}^m$ and high-frequency feature $\mathbf{X}_{\text{high}}^m$.

**(1) Spatio-temporal domain to frequency domain.** To decouple the original features, we first use Fourier transform to convert them into frequency domain, *i.e.*,

$$\mathbf{X}_f^m = \mathbf{DFT}(\mathbf{X}^m), \tag{1}$$

where $\mathbf{X}_f^m$ represents the corresponding Fourier-transformed feature in complex frequency domain.

**(2) High-pass and low-pass filtering.** In the frequency domain, we decompose $\mathbf{X}_f^m$ into different frequency components by designing a low-pass filter $\mathbf{M}_{\text{low}}$ and a high-pass filter $\mathbf{M}_{\text{high}}$. Then the low-frequency part $\mathbf{X}_{\text{low},f}^m$ and the high-frequency part $\mathbf{X}_{\text{high},f}^m$ are computed as follows:

$$\mathbf{X}_{f,\text{low}}^m = \mathbf{X}_f^m \cdot \mathbf{M}_{\text{low}}, \quad \mathbf{X}_{f,\text{high}}^m = \mathbf{X}_f^m \cdot \mathbf{M}_{\text{high}}. \tag{2}$$

**(3) Feature reconstruction with inverse Fourier transform.** To obtain the reconstructed low-frequency and high-frequency features, we apply the Inverse Discrete Fourier Transform (IDFT) to transform the low-frequency and high-frequency components from the frequency domain back to the spatio-temporal domain, then we can obtain the decoupled features as

$$\mathbf{X}_{\text{low}}^m = \mathbf{IDFT}(\mathbf{X}_{f,\text{low}}^m), \quad \mathbf{X}_{\text{high}}^m = \mathbf{IDFT}(\mathbf{X}_{f,\text{high}}^m). \tag{3}$$

The next task is to pinpoint the most suitable design of distillation loss for each type of features, respectively. As analyzed in Section 3.1, low-frequency features primarily encompass modality-generic information, highly shared across different modalities. Hence, it is imperative to maintain "strong consistency" for low-frequency features across different modalities so that their generality can be guaranteed. On the other hand, high-frequency features tend to capture modality-specific fine-grained information and are often accompanied by more noises. To preserve modality-specific details whilst reducing sensitivity to large errors stemming from the noises, we only require "weak consistency" for high-frequency features across different modalities.

As a result, for the low-frequency features on two different modalities $a$ and $b$, we use the conventional mean square error (MSE) as the loss function, *i.e.*,

$$\mathcal{L}_{\text{low}} = \frac{1}{ND} \left\| \mathbf{X}_{\text{low}}^a - \mathbf{X}_{\text{low}}^b \right\|^2, \tag{4}$$

where $N$ and $D$ denote the batch size and dimension, respectively.

While for the distillation of high-frequency features, a proper way is suppressing the significant gradient values caused by the noises and abnormally-large features. To this end, we leverage log mean square error (LogMSE) as the distillation loss, which has smoother gradients when the difference of

two feature values is large, as shown in Figure 4. The distillation loss for high-frequency features is formulated as

$$\mathcal{L}_{\text{high}} = \frac{1}{ND} \left\| \sigma(\mathbf{X}_{\text{high}}^a) - \sigma(\mathbf{X}_{\text{high}}^b) \right\|^2 \tag{5}$$

$$\text{with} \quad \sigma(\mathbf{X}) = \begin{cases} \log(1 + \mathbf{X}), & \mathbf{X} \geq 0 \\ -\log(1 - \mathbf{X}), & \mathbf{X} < 0 \end{cases}, \tag{6}$$

where $N$ and $D$ denotes the batch size and dimension, respectively.

## 4.2 ALIGNMENT OF FEATURE SCALE AND FEATURE SPACE

The consistency of feature distributions is pivotal for knowledge transfer. However, the significant differences in feature distributions between different modalities result in poor performance in cross-modal knowledge transfer. To mitigate the distribution discrepancies between modalities, we propose solutions from both the feature scale and feature space perspectives.

**(1) Feature scale alignment.** The inconsistency in feature scales is typically reflected in the fact that feature vectors from different modalities may have varying numerical ranges, which can negatively impact the effectiveness of knowledge distillation. To achieve feature scale alignment, we employed a

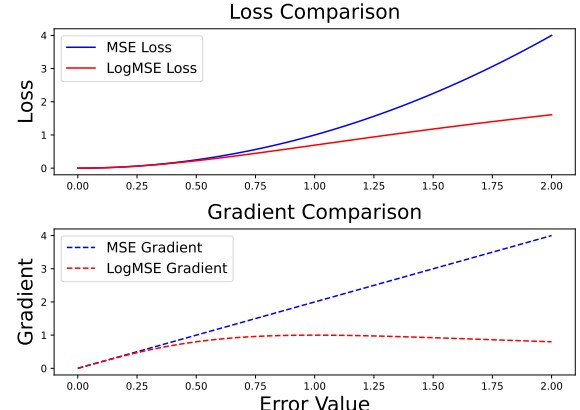

Figure 4: Comparison of loss and gradient between MSE and logMSE.

Feature Standardization strategy, which includes the following steps:

1. Mean Subtraction: First, mean subtraction is applied to the feature vectors to ensure that the mean of the features is zero, eliminating any bias in the features.

2. L2 Normalization: Next, L2 normalization is performed on the zero-centered feature vectors to ensure that the L2 norm of each feature vector is 1. This ensures that all feature vectors are compared on the same scale, avoiding computational biases caused by differences in the lengths of the feature vectors.

Herein is the formula for feature standardization:

$$\mathbf{Std}(\mathbf{X}) = \frac{\mathbf{X} - \bar{\mathbf{X}}}{\|\mathbf{X} - \bar{\mathbf{X}}\|_2}, \tag{7}$$

where $\mathbf{X}$ represents the input feature vector, $\bar{\mathbf{X}}$ represents the mean of the features, and $\|\cdot\|_2$ denotes the L2 norm. In practice, the mean subtraction operation can be directly implemented by using a DC filter in the frequency domain (as shown in Figure 3). By doing so, the previous distillation losses in Eq. 4 and Eq. 5 can be reformulated as follows:

$$\mathcal{L}_{\text{low}} = \frac{1}{ND} \left\| \mathbf{Std}(\mathbf{X}_{\text{low}}^a) - \mathbf{Std}(\mathbf{X}_{\text{low}}^b) \right\|^2, \tag{8}$$

$$\mathcal{L}_{\text{high}} = \frac{1}{ND} \left\| \sigma(\mathbf{Std}(\mathbf{X}_{\text{high}}^a)) - \sigma(\mathbf{Std}(\mathbf{X}_{\text{high}}^b)) \right\|^2. \tag{9}$$

**(2) Feature space alignment.** Although feature scale alignment can alleviate the inconsistency in the numerical ranges of features from different modalities, solely relying on scale alignment is insufficient to address the fundamental differences in feature distributions across modalities. Features from different modalities not only differ in numerical scales but may also exhibit significant variations in the specific shapes of their distributions and the delineation of class boundaries.

To further enhance the effective transfer of cross-modal knowledge, we propose an alignment strategy from the perspective of feature space, ensuring that the features of the teacher model and the

student model are comparable within the same space, thereby narrowing the distribution differences between modalities.

As shown in Figure 3, we designed an alignment module based on a shared classifier to achieve feature space alignment. Through the shared classifier, the features of both the teacher model and the student model can be aligned within the same decision space, thus reducing the distribution differences between modalities. Specifically, the features from the teacher model and the student model are fed into the same shared classifier, where they are classified through the shared classifier, and the classification alignment loss is defined as follows:

$$\mathcal{L}_{\text{align}} = \mathbf{CE}(\mathbf{\Phi}_h(\mathbf{X}^a_{\text{high}}), y) + \mathbf{CE}(\mathbf{\Phi}_h(\mathbf{X}^b_{\text{high}}), y) + \mathbf{CE}(\Phi_l(\mathbf{X}^a_{\text{low}}), y) + \mathbf{CE}(\Phi_l(\mathbf{X}^b_{\text{low}}), y), \quad (10)$$

where $\mathbf{CE}(\cdot)$ denotes the cross-entropy loss, $\mathbf{\Phi}_h$ and $\mathbf{\Phi}_l$ represent the shared classifiers for high-frequency and low-frequency features, respectively, and $y$ denotes the ground truth labels.

**Overall loss function.** In addition to the aforementioned losses, we also compute the cross-entropy loss on the raw features, low-frequency features, and high-frequency features of the student model, respectively, to ensure that these features are discriminative (Mao et al., 2023). We denote this loss as $\mathcal{L}_{\text{task}}$. See Figure 3 for an easier reference of all the losses we conduct. As a result, the total loss function can be expressed as follows:

$$\mathcal{L}_{\text{total}} = \mathcal{L}_{\text{task}} + \mathcal{L}_{\text{align}} + \lambda_1 \mathcal{L}_{\text{low}} + \lambda_2 \mathcal{L}_{\text{high}}, \quad (11)$$

where $\lambda_1$ and $\lambda_2$ represent the weighting parameters for the distillation losses of low-frequency and high-frequency features, respectively.

## 5 EXPERIMENTS

We evaluate our method on classification and semantic segmentation tasks across various multi-modal datasets. We provide experimental settings before detailing the result analysis.

### 5.1 CLASSIFICATION TASK

**Dataset.** CREMA-D (Cao et al., 2014) is an emotion recognition dataset with audio and vision, featuring six emotions: happy, sad, angry, fear, disgust, neutral. AVE (Tian et al., 2018) is an audio-visual event localization dataset with $4,143$ videos across 28 event categories. While VGGSound (Chen et al., 2020) is a large-scale audio-visual dataset with 210K ten-second videos, a subset of 50 categories for our experiments. CrisisMMD (Alam et al., 2018) is a multimodal dataset for natural disaster research, including annotated tweets and images from Twitter in image and text formats. For more detailed information about the dataset, please refer to Appendix A.

**Experimental Settings.** Our experimental settings follow (Huo et al., 2024; Fan et al., 2024; Wei et al., 2024). We use the ResNet-18 (He et al., 2016) as the backbone for audio-visual datasets and train them for 100 epochs in total. In the CrisisMMD dataset, we employ BERT-base (Devlin, 2018) and MobileNetV2 (Sandler et al., 2018) to extract text and visual features, respectively. We only train text modality for 20 epochs. We utilize the SGD optimizer with a momentum of $0.9$, and the batch size for training is set to $64$. For detailed training information, see Appendix A.

**Results Analysis.** In Table 1, we present the performance of our method on classification benchmarks. We compare the logit-based, feature-based, and cross-modal state-of-the-art distillation methods. that our proposed method consistently achieves the best performance across all datasets and modalities. For example, on the AVE dataset's visual modality, our method improves performance by 9%, reaching 47.8%, compared to the unimodal baseline. This highlights the effectiveness of our approach in transferring knowledge across modalities. Notably, our method excels in transferring knowledge from low-performing modalities to high-performing ones, where other methods fail. For instance, on CREMA-D's visual modality, AVE's audio modality, and VGGSound's audio modality, most methods underperform compared to the unimodal baseline, while our approach consistently improves performance by effectively transferring knowledge from weaker modalities. Additionally, our method is stable in bidirectional cross-modal transfer. On CrisisMMD, while DKD works well for text but not visual, and AFD succeeds for visual but fails for text, our method performs consistently across both modalities, achieving 79.1% on text and 72.7% on visual. This

Table 1: **The comparison of methods on Audio-Visual and Image-Text classification tasks.** The metric is the top-1 accuracy(%). 'A', 'V', and 'T' represent Audio, Visual, and Text modalities, respectively. "Uni" refers to unimodal models without distillation. "Logit" and "Feat" correspond to the original logit-based and feature-based distillation methods. "NKD" denotes the method where only Non-target Class logits are used for distillation. "C2KD" represents the cross-modal distillation method mentioned in (Huo et al., 2024). The best is in **bold**, and the second is underlined.

| Category | Method | CREMA-D | | AVE | | VGGSound | | CrisisMMD | |
|---|---|---|---|---|---|---|---|---|---|
| | | A | V | A | V | A | V | T | V |
| Uni-Modal | w/o KD | 62.4 | 66.8 | 63.7 | 38.8 | 68.9 | 44.9 | 77.4 | 70.2 |
| Logits | Logit | 61.7 | 62.6 | 60.0 | 39.1 | 65.7 | 45.4 | 78.5 | 70.5 |
| | DIST | 62.2 | 64.0 | 62.4 | 40.3 | 66.4 | 45.5 | 78.3 | 71.3 |
| | DML | 52.7 | 61.2 | 60.2 | 43.3 | 57.8 | 44.9 | 78.2 | 71.2 |
| | NKD | 62.4 | 61.8 | 60.7 | 38.1 | 65.6 | 44.9 | 78.1 | 71.2 |
| | DKD | 61.0 | 61.4 | 60.5 | 38.1 | 64.4 | 44.5 | 79.0 | 70.7 |
| Feature | Feat | 60.9 | 64.3 | 58.7 | 39.6 | 67.7 | 45.5 | 77.7 | 70.8 |
| | PKD | 60.4 | 64.8 | 58.0 | 41.0 | 62.9 | 46.9 | 77.5 | 70.9 |
| | OFD | 60.6 | 61.6 | 58.0 | 39.6 | 68.5 | 45.8 | 78.1 | 71.2 |
| | AFD | 61.2 | 59.5 | 62.7 | 38.8 | 68.7 | 45.8 | 69.8 | 72.3 |
| Cross-Modal | C2KD | 57.5 | 59.8 | 62.7 | 39.3 | 67.0 | 47.9 | 77.9 | 71.4 |
| | Ours | **64.1** | **71.0** | **64.9** | **47.8** | **70.0** | **48.1** | **79.1** | **72.7** |

Table 2: **The comparison of methods on semantic segmentation task.** The metric denotes the mean Intersect ion over Union(mIoU:%).

| Method | Uni | Logit | DIST | DKD | Feat | PKD | AFD | C2KD | Ours |
|---|---|---|---|---|---|---|---|---|---|
| Depth | 30.9 | 29.7 | 32.3 | 32.5 | 29.4 | 31.0 | 30.2 | 31.8 | **33.2** |
| RGB | 34.1 | 32.8 | 34.9 | 35.3 | 32.8 | 33.7 | 32.7 | 34.8 | **36.9** |

outstanding performance is attributed to our method's ability to capture both modality-specific and modality-agnostic information through frequency decomposition and customized loss functions, as well as mitigating inherent feature distribution differences through feature alignment. This ensures robust results across various modality pairs (A-V, T-V) and network architectures (ResNet-ResNet, BERT-MobileNet).

## 5.2 SEMANTIC SEGMENTATION TASK

**Dataset.** NYU-Depth V2 (Wofk et al., 2019) is a multimodal dataset for indoor scene understanding research. It provides two modalities of depth information and RGB image information. There are a total of 40 categories. It contains 1, 449 densely labeled RGB and depth image alignment pairs.

**Experimental Settings.** Following C2KD (Huo et al., 2024), the DeepLab V3+ (Chen et al., 2018) model is utilized with ResNet-18 as the backbone, which is initialized with the pre-trained weights on ImageNet (Deng et al., 2009). We train the student for 50 epochs in total and the batch size is 16.

**Results Analysis.** Regarding segmentation task, Table 2 shows the performance of various KD methods on NYU-Depth V2. Our method still consistently outperforms all other methods, with 33.2% mIoU for Depth and 36.9% mIoU for RGB. These results surpass the next best method (DIST for Depth, DKD for RGB) by a notable margin of 0.9% and 1.6%, respectively. As highlighted in Section 5.1 for classification tasks, our method is also stable in bidirectional cross-modal transfer in segmentation tasks.

## 6 ANALYSIS

In this section, we first evaluate the effectiveness of the key components in our CMKD method, including frequency decomposition, feature alignment, and loss functions, through ablation studies. We then conduct sensitivity analyses on frequency loss and loss coefficients to ascertain their impact on performance. Finally, we visualize the feature distributions using t-SNE, showcasing the improved feature separation and cross-modal knowledge transfer achieved by our proposed method, when compared with traditional techniques.

### 6.1 EFFECTIVENESS OF COMPONENTS IN CMKD

We perform experiments to show the effectiveness of each proposed component in CMKD in Table 3. Firstly, it is evident that each individual component contributes positively to the overall performance. **Frequency decomposition distillation** provides improvements for most modalities as it helps to separate modality-specific information from modality-generic information. However, these improvements are not always consistent; for example, on the audio (A) modality of the CREMA-D dataset, there is a 0.1% performance drop. This inconsistency may stem from significant differences in feature distributions across modalities. When we add the **Feature space alignment** and **Feature standardization** modules, the cross-modal performance improves significantly, highlighting the importance of reducing feature distribution discrepancies between modalities. Moreover, applying **logMSE loss** to high-frequency features enhances the transmission of modality-specific information, indicating that it is not necessary to fully align modality-specific information, and maintaining a weak consistency is more effective for transferring such information. Finally, the comprehensive integration of all components ensures more robust cross-modal knowledge transfer, thereby achieving more stable performance across different modalities.

Table 3: **The ablation analysis of different components.** Freq represents frequency decomposition, Align refers to the feature space alignment module, Scale indicates feature standardization for scale alignment, and Log represents the use of logMSE loss on high-frequency features. Baseline refers to the original feature distillation.

| Method | | | | CREMAD | | AVE | |
|---|---|---|---|---|---|---|---|
| Freq | Align | Scale | Log | A | V | A | V |
| | | | | 60.9 | 64.3 | 58.7 | 39.6 |
| ✓ | | | | 60.8 | 68.7 | 61.0 | 43.3 |
| | ✓ | | | 60.9 | 67.9 | 63.2 | 41.3 |
| ✓ | ✓ | | | 61.8 | 68.7 | 62.4 | 45.8 |
| ✓ | | ✓ | | 62.2 | 70.0 | 62.4 | 44.8 |
| ✓ | ✓ | ✓ | | 62.2 | 70.6 | 62.4 | 46.0 |
| ✓ | ✓ | ✓ | ✓ | 64.1 | 71.0 | 64.9 | 47.8 |

### 6.2 SENSITIVITY STUDY OF FREQUENCY

In this paper, we employ high-frequency and low-frequency features to imitate the teacher model, respectively. In this subsection, we conduct experiments on high-frequency loss and low-frequency loss to investigate their influences on performance. As shown in Table 4, both the high-frequency loss and low-frequency loss lead to significant accuracy improvements. Furthermore, considering the frequency with character, we find high-frequency benefits more to the audio modality and low-frequency benefits more to the visual modality. Besides, when combining the high-frequency loss and low-frequency loss, we achieve the best performance, which indicates that transferring modality-generic information and modality-specific information can complement each other effectively to enhance the overall performance.

Table 4: **The analysis of frequency loss.**

| Method | | CREMAD | | AVE | |
|---|---|---|---|---|---|
| High | Low | A | V | A | V |
| ✓ | | 62.4 | 67.5 | 64.9 | 44.0 |
| | ✓ | 62.1 | 69.1 | 64.2 | 44.5 |
| ✓ | ✓ | 64.1 | 71.0 | 64.9 | 47.8 |

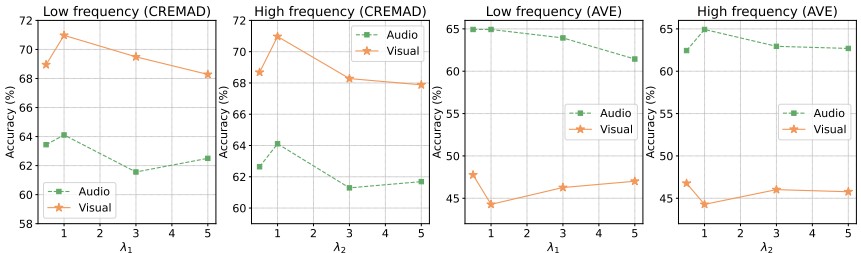

Figure 5: **Sensitivity study of high and low-frequency loss weight.**

## 6.3 SENSITIVITY STUDY OF LOSS COEFFICIENTS

In Eq. 11, we utilize the coefficients $\lambda_1$ and $\lambda_2$ to balance the low-frequency and high-frequency, respectively. In this subsection, we do the sensitivity study of loss weight on CREMA-D and AVE. The results are shown in Figure 5. We conducted experiments by setting the weights to 0.5, 1, 3, and 5, respectively. On the CREMA-D dataset, both modalities achieve the best performance when $\lambda_1 = 1$ and $\lambda_2 = 1$. On the AVE dataset, the audio modality performs best with $\lambda_1 = 1$ and $\lambda_2 = 1$, while the visual modality achieves the highest accuracy with $\lambda_1 = 0.5$ and $\lambda_2 = 0.5$. Therefore, $\lambda_1 = 1$ and $\lambda_2 = 1$ generally represent a good configuration.

## 6.4 VISUALIZATION

In Figure 6, we present a t-SNE (Van der Maaten & Hinton, 2008) visualization to compare the performance of the original feature distillation method and our proposed approach in cross-modal knowledge distillation. Figure 6(a) illustrates the result of traditional feature distillation, where there is significant overlap between the features of the visual modality (teacher) and audio modality (student).

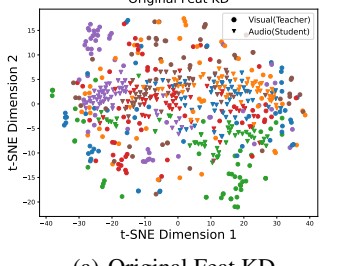
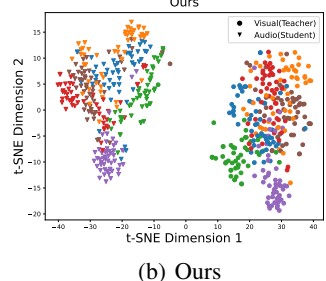

(a) Original Feat KD      (b) Ours

Figure 6: **t-SNE visualization comparison between the conventional feature distillation method and our proposed approach.** We selected six classes from the VGGSound testset for visualization.

This indicates that the method fails to effectively differentiate between modality-specific features, leading to suboptimal retention of modality-specific information and reduced discriminative capacity in the student model. In contrast, Figure 6(b) demonstrates the visualization of our approach, where the features from different modalities are clearly separated, forming two distinct clusters. This separation suggests that our method successfully disentangles modality-generic and modality-specific information, improving feature discrimination. These results validate our frequency decomposition strategy, which preserves modality-specific characteristics while enhancing cross-modal knowledge transfer.

## 7 CONCLUSION

In this paper, we investigated the non-negligible challenges faced cross-modal knowledge distillation, particularly focusing on the discrepancies between modality-specific and modality-generic information, and the differences in feature distributions across modalities. Based on the experimental observation that low-frequency features exhibit stronger correlations than high-frequency features, we proposed a novel distillation framework – It decouples these types of information through frequency-based feature analysis and introduces a differentiated distillation strategy for different frequency components. Additionally, we addressed feature distribution discrepancies by incorporating a scale consistency loss and using a shared classifier for feature space alignment. Extensive experiments have demonstrated the effectiveness of our approach.

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

# A EXPERIMENTAL SETUP

## A.1 DATASET

**CREMA-D** (Cao et al., 2014) is a dataset for emotion recognition research, including two modalities of audio and vision. The dataset includes six emotion categories: happy, sad, angry, fear, disgust, and neutral. It contains 7,442 video clips, of which 6,698 are used as the training set and 744 are used as the test set.

**AVE** (Tian et al., 2018) is a dataset for audio-visual event localization, including two modalities of audio and vision. It contains a total of 4,143 videos, covering 28 event categories. The division of the training set, validation set, and test set refers to (Tian et al., 2018).

**VGGSound** (Chen et al., 2020) is a large-scale audio-visual dataset consisting of more than 210,000 10-second videos. We randomly selected a dataset composed of 50 categories for experiments. In total, it includes 32,496 videos, of which 29,999 are divided into the training set and 2,497 are divided into the test set.

**CrisisMMD** (Alam et al., 2018) is a multimodal dataset for research related to natural disasters. It consists of manually annotated tweets and pictures from Twitter, including two modalities of image and text. In total, it includes five categories (rescue, not humanitarian, affected individuals, infrastructure and utility damage, other relevant information) and XX pieces of data. The division of the training set, validation set, and test set refers to (Alam et al., 2018).

**NYU-Depth V2** (Wofk et al., 2019) is a multimodal dataset for indoor scene understanding research. It provides two modalities of depth information and RGB image information. There are a total of 40 categories. It contains 1,449 densely labeled RGB and depth image alignment pairs. Among them, 795 are divided into the training set and 654 are divided into the test set.

## A.2 DATA PREPROCESSING DETAILS

We follow (Huo et al., 2024; Fan et al., 2024; Wei et al., 2024) and provide our preprocessing details. For the audio-visual datasets, the audio data is converted into spectrograms with a size of 257×299 for CREMA-D and 257×1,004 for both AVE and VGGSound. The spectrograms are generated using a window length of 512 and an overlap of 353. For visual modality, during training, 1 frame is extracted from AVE and CREMA-D, while 3 frames are uniformly sampled from VGGSound. During testing, the middle frame is selected. Random cropping and flipping data augmentation methods were applied during training, and the same approach was used for the visual modality in the CrisisMMD dataset. For the NYU V2 dataset, we applied random HSV and random flipping as data augmentation techniques on the RGB modality.

## A.3 NETWORK ARCHITECTURES

For the audio-visual datasets, we use ResNet-18 as the backbone network. In the CrisisMMD dataset, we use the pre-trained BERT-base as the backbone for text and MobileNetV2 as the backbone for images. For the NYU V2 dataset, we use DeeplabV3+ with a ResNet-18 backbone as the network architecture. Additionally, during distillation, we use the features extracted before the ReLU activation. Since the core of segmentation tasks is to generate pixel-level classification results rather than mapping global features to a fixed class, our method does not use a shared classifier alignment module for segmentation tasks.

## A.4 TRAINING DETAILS

**Optimizer:** For BERT, we use the Adam optimizer, while for the others, we use the SGD optimizer with a momentum of 0.9.

**Learning rate:** For BERT, a fixed learning rate of 1e-5 is used. For segmentation tasks, the initial learning rate is 0.02 and decays according to the 'poly' policy with a power of 0.9. For all other tasks, the initial learning rate is set to 1e-2 and follows the 'poly' decay policy with a power of 0.9.

**Batch size:** For segmentation tasks, the batch size is 16, while for all other tasks, it is set to 64.

**Epochs:** For BERT, since it is pre-trained, we only train for 20 epochs. For segmentation tasks, we train for 50 epochs, and for all other tasks, we train for 100 epochs.

## A.5 TRAINING ENVIRONMENT

All experiments were conducted on NVIDIA Tesla V100 and RTX 3090 GPUs using CUDA 12.5 with the PyTorch framework.

# B CALCULATION OF CKA SCORE

Centered Kernel Alignment (CKA) is a metric used to measure the similarity between two sets of feature representations, often from different layers or models. It helps in understanding how neural networks encode data.

CKA compares two sets of features by aligning their Gram matrices, which capture pairwise similarities between samples. It has the following properties:

- **Invariant to orthogonal transformations**: Features can be compared even if they are in different spaces.
- **Scale invariant**: The score is unaffected by differences in feature magnitudes.

Given two sets of feature representations $\mathbf{X} \in \mathbb{R}^{n \times p}$ and $\mathbf{Y} \in \mathbb{R}^{n \times q}$, where $n$ is the number of samples, the steps are as follows:

1. **Gram matrix**:
$$\mathbf{K_X} = \mathbf{X}\mathbf{X}^\top, \quad \mathbf{K_Y} = \mathbf{Y}\mathbf{Y}^\top$$

2. **Centering the Gram matrices**:
$$\tilde{\mathbf{K_X}} = \mathbf{K_X} - \frac{1}{n}\mathbf{1}\mathbf{K_X} - \frac{1}{n}\mathbf{K_X}\mathbf{1} + \frac{1}{n^2}\mathbf{1}\mathbf{K_X}\mathbf{1}$$

3. **CKA Score**:
$$\mathrm{CKA}(\mathbf{X}, \mathbf{Y}) = \frac{\mathrm{Tr}(\tilde{\mathbf{K_X}}\tilde{\mathbf{K_Y}})}{\sqrt{\mathrm{Tr}(\tilde{\mathbf{K_X}}\tilde{\mathbf{K_X}})\mathrm{Tr}(\tilde{\mathbf{K_Y}}\tilde{\mathbf{K_Y}})}}$$

The CKA score ranges from 0 to 1. A score close to 1 indicates high similarity between the two sets of features, while a score near 0 indicates low similarity.

