# OpenReview forum: "Frequency-Decoupled Cross-Modal Knowledge Distillation"
_ICLR.cc/2025/Conference — ICLR 2025 Conference Withdrawn Submission_

### Official Review · Reviewer_GeLW · 2024-11-01

**Soundness:** 2
**Presentation:** 2
**Contribution:** 2
**Rating:** 5
**Confidence:** 3

**Summary:**

The authors start this paper with an interesting observation: When distilling in multi-modality setting, low-frequency features tend to provide information which is modality-generic, while high-frequency features tend to provide the information that is more modality-specific. While this is unclear why it happens, the authors also justify this by showing that the first and second moment of audio modality features are quite a bit higher than those of vision features.

Thus, the authors, first decouple the features using Fourier transforms into high and low-frequency ones, and then treat them differently by using MSE loss for the low-frequency ones and some logarithmic version of it for the high-frequency ones. Furthermore, they also provide some alignment module for better performance.

The authors show good results in a wide range of datasets, and provide adequate ablation studies to show the performance improvement of each module.

**Strengths:**

I liked the following aspects of the paper:

1) The observation about the low and high-frequency features in cross-domain distillation is interesting and the first time I see in the literature. Then the authors decouple features into low and high-frequency using Fourier transforms, which makes absolutely sense, and treat them differently using different losses.

2) The results seem quite good, and in different datasets the authors are getting a few points of improvement. Furthermore, they show results in different settings: Audio-Visual and Image-Text classification, in addition to semantic segmentation tasks, making the evaluation quite extensive.

3) The paper does adequate ablation and analysis, with the most important one being the ablation on the different components of the proposed model.

**Weaknesses:**

I think the paper might have the following points that can be improved:

1) Unclear motivation for the usage of logarithmic MSE. I understand that the loss seems to work, but it is unclear to me why using logarithmic MSE comes with an improvement in performance when used for high-frequency domain.

Might be interesting to see a couple of alternatives to it:

a) Normalize the features of both teacher and student before applying some standard feature distillation loss, so they are in the same scale.

b) Weight the high-frequency domain loss less than the low-frequency ones. With logarithmic being a monotonic function, it looks to me that it is similar to weighting the loss down, so would be interesting to see if this is really the case.

2) It is unclear to me the need and justification for the feature alignment module (section 4.2). Furthermore, it is very unclear if this module is adding anything.

The authors are required to expand more in the motivation for it, and are asked to do an ablation, where Frequency, Scale and Log are turned on but the Align is turned off, to see the real effect of it.

3) Some parts in writing seem very basic. The people who are going to read this paper are most likely experts in the field, so not sure there is a need to write equations on what is MSE loss, feature standardization, or doing them all over again with an applied normalization. Pretty much everyone who will ever read this paper knows those things, thus equations 4, 7, 8, 9 can be safely removed, they definitely do not make the paper more 'mathematical'.

**Questions:**

1) Please provide the alternative experiments to the logarithmic loss I mentioned in weakness (1).

2) Please provide the experiment I asked in weakness (2).

3) Please explain how are the M_low and M_high filters designed (this is completely lacking from the paper).

I am initially scoring the paper as Borderline Reject but will increase the score if the authors are able to provide good responses in Weaknesses and Questions part.

---

### Official Review · Reviewer_uFFt · 2024-11-03

**Soundness:** 2
**Presentation:** 2
**Contribution:** 2
**Rating:** 5
**Confidence:** 4

**Summary:**

The paper focuses on cross-modality knowledge distillation (CMKD). It first observes that "low-frequency features tend to capture modality-agnostic, generalizable information, while high-frequency features are more modality-specific." It then proposes frequency-decoupled cross-modal knowledge distillation, a method designed to decouple and balance knowledge transfer across modalities by leveraging frequency-domain features. It conducts experiments on multiple datasets spanning audio-visual, text-visual, and depth-RGB modalities. Results show that the proposed methods even outperforms single-modality knowledge distillation on all datasets.

**Strengths:**

- It is reasonable to explore pretrained models on a different yet related modality in knowledge distillation, although it is too harsh to only allow cross-modal distillation instead of allowing using models on both modalities in knowledge distillation.

- Datasets in experiments are good that cover diverse modalities and tasks.

- It is fresh to study the correlation between frequency/spectrum and modality, although it does not sufficiently demonstrate whether there is a strong correlation.

**Weaknesses:**

- The paper uses CKA scores to demonstrate that "the low-frequency obtains the highest scores, while the high-frequency are more dissimilar than the original features" (Fig. 1). The paper admits that "all scores are small due to modality and model differences". Even so, it is not clear whether the small CKA scores and the comparisons of CKA between features of different modalities are statistically significant. It is hard to understand whether such small CKA differences can yield significant boost in knowledge distillation as shown in Table 1.

- While the paper motivates the work by stating "low-frequency features tend to capture modality-agnostic, generalizable information, while high-frequency features are more modality-specific", it is better to show some visual examples to better understand the high/low-frequency features and qualitatively understand how they are correlated to different modalities.

- The paper assumes that the readers have rich background of cross-modality knowledge distillation (CMKD). In fact, the reviewer is confused why it only allows one to do knowledge distillation over a model trained on a different modality. Is it too harsh and unnecessarily difficult? In fact, the paper also mentions "vision-to-language distillation", which naturally reminds the reviewer think about a setting that allows using both text encoder and visual encoder in knowledge distillation.

- It is not clear whether cross-modality knowledge distillation always helps train a small student model, and when using cross-modal teacher model helps or hurts. Shall one assume that uni-modal knowledge distillation is a kind of "upper bound" as cross-modal features might not be comparable to the modality of interest?

**Questions:**

Authors are encouraged to address the weaknesses in rebuttal/responses. Please refer to the weaknesses for details.

---

### Official Review · Reviewer_jG7G · 2024-11-03

**Soundness:** 2
**Presentation:** 3
**Contribution:** 2
**Rating:** 3
**Confidence:** 4

**Summary:**

This paper introduces a technique for cross-modal distillation. The authors argue that different modalities capture generic and specific information and propose to decouple them using Fourier transformation. Specifically, they claim that low-frequency feature carries modality-generic information and is favorable for cross-modal distillation. They also propose the *feature scale alignment* and *feature space alignment* as auxiliary loss. The authors validate their method on classification and semantic segmentation tasks.

**Strengths:**

1. Modality imbalance is a well-known problem for cross-modal learning. The idea of decoupling the feature into modality-generic and modality-specific information is interesting.
2. The paper is well-written and easy to follow.
3. The authors provide diverse and extensive experiments on different datasets.

**Weaknesses:**

1. The authors introduce the motivation in Figure 1 and line 72: *As illustrated in Figure 1, low-frequency features obtain the
highest CKA scores, indicating that they contain more modality-generic information. In contrast,
high-frequency features show lower CKA ...* However, in both datasets, the CKA scores of low-freq feature are just slighly larger than that of the original feature, e.g., 0.3 v.s. 0.28 in AVE. In this case, the claim that low-freq feature capture modality-generic information is not convincing.
2. The proposed *feature space alignment* is similar to [1], where teacher shares the classifier with student.
3. I have some concerns about the experimental results. The authors claim that **Our experimental settings follow (Huo et al., 2024, ...* (line 361). However, the results in Table 1\& 2 are different from the results of Table 3 \& 4 in [2].
4. In Table 3, it seems like the proposed freqency decomposition could be harmful (e.g. 60.9 $\rightarrow$ 60.8 in CREMAD with audio modality). In addition, without LogMSE, the model achieves the same result with DIST (see Table 1). However, LogMSE is a common trick in deep learning.

[1] Knowledge distillation via softmax regression representation learning. ICLR 2021

[2] C2KD: Bridging the Modality Gap for Cross-Modal Knowledge Distillation. CVPR 2024

**Questions:**

1. Table 1 shows that visual modality is favorable in CREMA-D while audio modality is dominant in AVE and VGGSound. I think the discussion about modality bias in different scenarios can bring more insights into cross-modal distillation.

---

### Official Review · Reviewer_F8pm · 2024-11-03

**Soundness:** 3
**Presentation:** 3
**Contribution:** 3
**Rating:** 5
**Confidence:** 4

**Summary:**

In this paper, the author presents a cross-modal knowledge distillation technique, which integrates both low-frequency and high-frequency features for efficient cross-modal knowledge distillation. The proposed technique is validated in various experimental settings, resulting in significant improvement.

**Strengths:**

1) The proposed method achieves significant enhancement.
2) The author introduced a novel knowledge distillation technique that combines low-frequency and high-frequency features for efficient cross-modal knowledge distillation.
3) The author conducted a series of experiments under various configurations.

**Weaknesses:**

The core contribution is similar to some existing works [1-2]. The main differences between the proposed method and some high related works [1-2] are not given.

References:
[1] FreeKD: Knowledge Distillation via Semantic Frequency Prompt
[2]  Extracting Low-/High- Frequency Knowledge from Graph Neural Networks and Injecting it into MLPs: An Effective GNN-to-MLP Distillation Framework

**Questions:**

The main differences between frequence-related methods [1,2] should be given.

[1] FreeKD: Knowledge Distillation via Semantic Frequency Prompt
[2]  Extracting Low-/High- Frequency Knowledge from Graph Neural Networks and Injecting it into MLPs: An Effective GNN-to-MLP Distillation Framework.

---

### Note · Authors · 2024-11-13

I have read and agree with the venue's withdrawal policy on behalf of myself and my co-authors.